# Evolving a 24-hr oscillator in budding yeast

**Gregg A Wildenberg[1,2]\*, Andrew W Murray[1,2]**

[1]Faculty of Arts and Sciences Center for Systems Biology, Harvard University, Cambridge, United States; [2]Department of Molecular and Cellular Biology, Harvard University, Cambridge, United States

**Abstract** We asked how a new, complex trait evolves by selecting for diurnal oscillations in the budding yeast, *Saccharomyces cerevisiae*. We expressed yellow fluorescent protein (YFP) from a yeast promoter and selected for a regular alternation between low and high fluorescence over a 24-hr period. This selection produced changes in cell adhesion rather than YFP expression: clonal populations oscillated between single cells and multicellular clumps. The oscillations are not a response to environmental cues and continue for at least three cycles in a constant environment. We identified eight putative causative mutations in one clone and recreated the evolved phenotype in the ancestral strain. The mutated genes lack obvious relationships to each other, but multiple lineages change from the haploid to the diploid pattern of gene expression. We show that a novel, complex phenotype can evolve by small sets of mutations in genes whose molecular functions appear to be unrelated to each other.

## Introduction

Biological oscillators demand dynamical interactions between multiple components and have evolved many times to drive various behaviors, from heartbeats, on the timescale of seconds, to circadian clocks and seasonal clocks, on the time scale of days and years. The budding yeast, *Saccharomyces cerevisiae*, lacks homologs to genes that make up the core oscillator or light-sensing pathways of fungal circadian rhythms (*Brunner and Kaldi, 2008*; *Idnurm and Heitman, 2010*; *Salichos and Rokas, 2010*) and does not display any 24-hr behavior capable of sustained autonomous oscillations in a constant environment (*Eelderink-Chen et al., 2010*; *Robertson et al., 2013*). As an example of a novel trait, we evolved a 24-hr oscillator in the budding yeast to determine how biologically complex machinery could evolve from proteins and signaling networks that lack any known connection to oscillatory behavior. Because circadian clocks can be entrained by environmental signals and have a temperature-independent period, whereas the behavior we have evolved represents an autonomous oscillator with a period of roughly 24 hr that controls a biological output, we refer to the circuit we have evolved as a diurnal oscillator, referring to its period, rather than the distinction between day and night.

## Results

Our initial goal was to use fluorescence activated cell sorting (FACS) and fluorescent reporters to select for diurnal regulation of gene expression. We expressed yellow fluorescent protein (YFP) from the promoter of the *FLO1* gene (*Verstrepen et al., 2005*; *Rando and Verstrepen, 2007*; *Verstrepen and Fink, 2009*) and elevated the mutation rate about 100-fold by mutating the proof-reading activity of DNA polymerase ∂ (*POL3*) (*Jin et al., 2005*). We used FACS to select for variation in YFP fluorescence over a 24-hr period: each cycle began with selection of the dimmest events, the selected cells were allowed to proliferate, and the population was then selected, 10 hr later; for the brightest events,

\*For correspondence: greggwildenberg@gmail.com

**Competing interests:** The authors declare that no competing interests exist.

**Reviewing editor**: Michael Laub, Massachusetts Institute of Technology, United States

**eLife digest** In living things, many important behaviors—including animal heartbeats and sleep patterns—happen in cycles. Machines called oscillators, which are found inside cells, control these behaviors. There are many different oscillators and they share some common features, despite involving different genes.

Each oscillator is formed of a set of genes that interact with each other to drive regular cycles lasting seconds, hours, or even months. The oscillators do not need any cues from the environment to maintain these cycles. However, cues such as light or temperature can keep the oscillator synchronized with the environment.

To ask how complex machines like oscillators could evolve, Wildenberg and Murray inserted a gene that makes a fluorescent protein into budding yeast, a single-celled species that does not have an oscillator with a period of 24 hr. These yeast cells were then selectively grown through a few hundred generations to experimentally evolve a yeast strain where the levels of protein fluorescence regularly alternated over 24-hr periods.

Wildenberg and Murray then carried out further experiments to discover the cause of the pattern of protein fluorescence. These revealed that the pattern was due to the yeast cells alternating between forming clumps of multiple cells and living separately. The genes that mutated to create the cycles of cell clumping in the yeast all appear to have unrelated roles.

The 24-hr oscillator that evolved in the yeast has some of the features of the biological oscillators found in nature. It maintains regular cycles even without any cues from the environment and it can control a cell behavior, but the oscillator appears to be unable to accept cues from the environment, a universal property of naturally evolved circadian clocks. Further work to understand how the genes work together in the oscillator will help to better understand how 24-hr oscillators in nature can evolve from genes that lack any 24-hr behavior.

the selected cells were allowed to proliferate, and the dimmest events were selected, 14 hr later, completing the cycle (*Figure 1A*). We evolved two parallel populations using two different strengths of selection: the top and bottom 6% (E6) or 15% (E15) of the events detected by FACS. The selected cells proliferated for roughly six divisions before the next selection. Our scheme selects for synchrony rather than entrainment: twice a day, we select the cells with high or low fluorescence, rather than varying an environmental cue and selecting for a response to it. The human analogy would be to go to a large, international airport and select only those travelers who had arrived from the same time zone, rather than taking all newly arrived passengers and using alternating light and dark cycles to entrain their clocks over a period of several days.

After 30 days of selection, the intensity distributions resulting from the selections for high and low fluorescence differed. To analyze the behavior of the evolved populations in more detail, we cultured them from stocks frozen that had been prepared at the end of the original selection. When they are first thawed, these populations are asynchronous, presumably because the manipulations associated with freezing and thawing disrupt the synchrony between the oscillations of different individuals. We therefore subjected these cultures to the same, twice daily, selection used for the evolution: the dimmest events are collected, cells are allowed to proliferate for 10 hr, the population distribution of fluorescence is recorded by FACS, and the brightest events are collected. After 14 hr of proliferation, distributions are recorded, and the dimmest events are collected to complete the cycle (*Figure 1B*). This protocol is a form of selection synchrony: it synchronizes a population by selecting those members that are at the peaks and troughs of a cycle of fluorescence intensity, rather than by applying a stimulus that alters the phase of the oscillations until all members of the population have the same phase (induction synchrony or entrainment). After synchronization, both evolved populations showed a cyclical change in the fluorescence distribution, with the E6 population giving larger shifts than the less strongly selected E15 (*Figure 1C*). It takes more cycles of synchronization of E15 to achieve the full amplitude of daily oscillation than it does for E6, which is largely synchronized within 24 hr of selection.

Our starting strain also contained red fluorescent protein (RFP) under the control of the *ACT1* (actin) promoter, allowing us to ask if RFP fluorescence, which had not been selected on, also cycled.

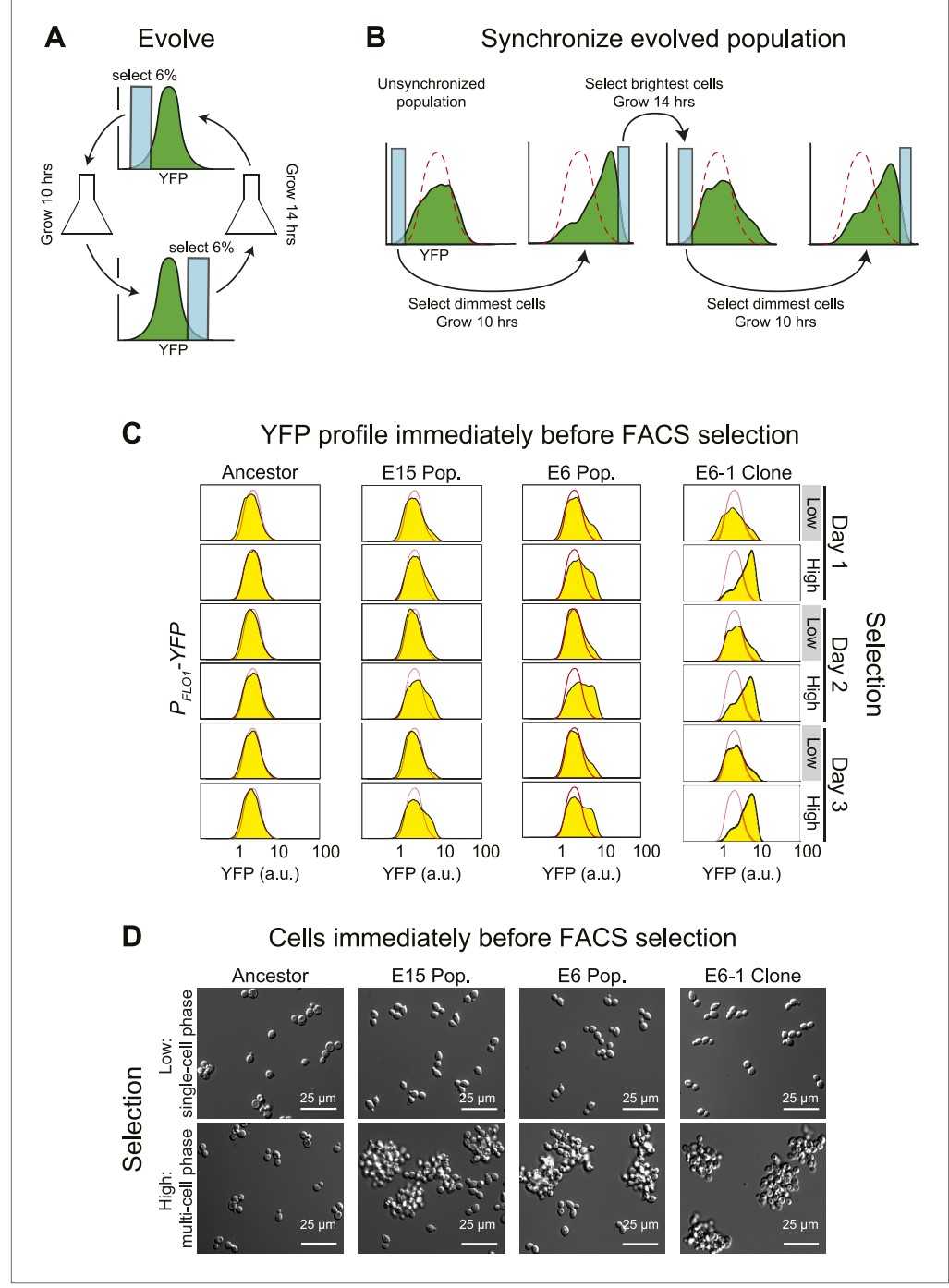

**Figure 1**. Cell association clock evolves from selecting for high and low YFP expression. (**A**) Selection scheme for a 24-hr oscillator: cells were collected from the dimmest 6% (E6) of the distribution of YFP fluorescing events, grown exponentially at <1 × 10⁶ cells/ml for ∼10 hr and then the brightest 6% of events were collected and the cells in this population were grown for ∼14 hr at <1 × 10⁶ cells/ml. The selection cycle was repeated for 30 days. The E15 population was evolved in the same manner, but the dimmest and brightest 15% of events were collected in the two phases of the selection. (**B**) Scheme for synchronizing YFP oscillations from an unsynchronized population: the dimmest events from an unsynchronized population (blue box) are collected, thus selecting for cells at the single cell phase of the oscillation. After 10 hr of growth, distributions are first recorded and the brightest YFP events (clump of cells) are collected, grown for 14 hr, distributions are recorded again and the dimmest events are selected to complete the cycle. This cycle is repeated to keep the population maximally synchronized. The dashed red lines

*Figure 1. Continued*

represent the fluorescence distribution of the ancestral population. (**C**) Representative density plots showing oscillations in YFP fluorescence of a FACS-synchronized sample of the E6 and E15 populations after terminating the selection, and a clone isolated from E6 (E6-1). The red outline overlay shows the average ancestral distribution. (**D**) Representative DIC images of synchronized populations immediately before FACS selection for low (single-cell phase) and high (multi-cell phase) YFP fluorescence.

The following figure supplement is available for figure 1:

**Figure supplement 1**. Evolution of 24-hr cell aggregation clock.

To our surprise, the distribution of RFP also cycled (*Figure 1—figure supplement 1A*), prompting us to examine the cells at different phases of the cycle. Just before their scheduled selection for low fluorescence, the population was predominantly single-celled, and just before their selection for high fluorescence, most cells were in multicellular clumps (*Figure 1D*). The fluorescence per cell is constant over 24 hr, for both YFP and RFP, and thus the cyclical variation in intensity seen by FACS is entirely due to changes in cell association (*Figure 1—figure supplement 1B*). Because we see similar fluctuations in the fluorescence from $P_{ACT1}$-*RFP* as we do from $P_{FLO1}$-*YFP*, we believe that we would have selected for oscillations in cell aggregation with any promoter that drove the expression of a fluorescent protein to levels that were sufficient for our FACS-based selection.

The cyclical behavior of our evolved populations could be due to interactions between two or more different genotypes. To eliminate this possibility, we isolated and analyzed fifteen clones from each population after subjecting them to FACS-based synchronization: all 30 clones produced strong oscillations in YFP and RFP distributions, and we analyzed the clone that showed the strongest oscillations (E6-1) in detail (*Figure 1C,D*). Except for the free-run experiments, described later in the paper, all other experiments examined populations that were subject to FACS-based selection twice a day with a 10-hr interval between the selection for the dimmest and brightest events and a 14-hr interval between the selection for the brightest and dimmest events.

The cycle is not a response to daily variations in environmental factors such as temperature. If populations were responding to environmental fluctuations, they should eventually synchronize in response to these fluctuations. To look for this behavior, we subdivided a single population of clone E6-1 and subjected it to two different treatments: synchronizing it by FACS selection, as described above, or subjecting it to identical manipulations, except that we collected all the events that passed through the FACS, without regard to their fluorescence intensity. The first population oscillated, showing primarily single cells just before the selection for low YFP fluorescence, and primarily clumps just before the selection for high YFP fluorescence, whereas the unsynchronized population showed very similar distributions at these two times (*Figure 1—figure supplement 1C, D*). We conclude that we have evolved a cyclical change in cell association with a period of about 24 hr, and cells that show this oscillation can be synchronized by selecting on their level of fluorescence as a proxy for the number of cells in a clump.

To analyze the oscillations in cell association, we followed the behavior of E6-1 cells grown at low population density using two different labels. The first, cerulean fluorescent protein (CFP) was expressed from a constitutive promoter ($P_{ACT1}$) to distinguish two genetically identical subclones: one expressed CFP (CFP+) and the other did not (CFP−). The second was introduced by labeling cells, just after FACS selection, by covalently linking Oregon Green 488-X, a yellow fluorescent molecule, to their cell walls (*Hoch et al., 2005*). Because a daughter's cell wall is entirely new, the original cells retain the yellow label and their daughters are unlabeled (*Barral et al., 2000*). We followed these cultures as they proliferated and eventually switched to the other phase of the morphological oscillation (*Figure 2A,B* and *Figure 2—figure supplement 1A*). Single cells that were selected for by sorting for low YFP fluorescence and covalently labeled (T = 0 hr), proliferated to become clumps composed of a single yellow cell surrounded with non-yellow cells that had the same CFP expression state as the original single cell. The absence of clumps containing more than one yellow cell or both CFP+ and CFP− cells in the same clump shows that cells emerging from the single-cell phase build lineage-based clumps. Consistent with this interpretation, we saw synchronous oscillations in cell association even at very high dilution (~$10^3$ cells/ml). Multicellular clumps, which were covalently labeled at the time of

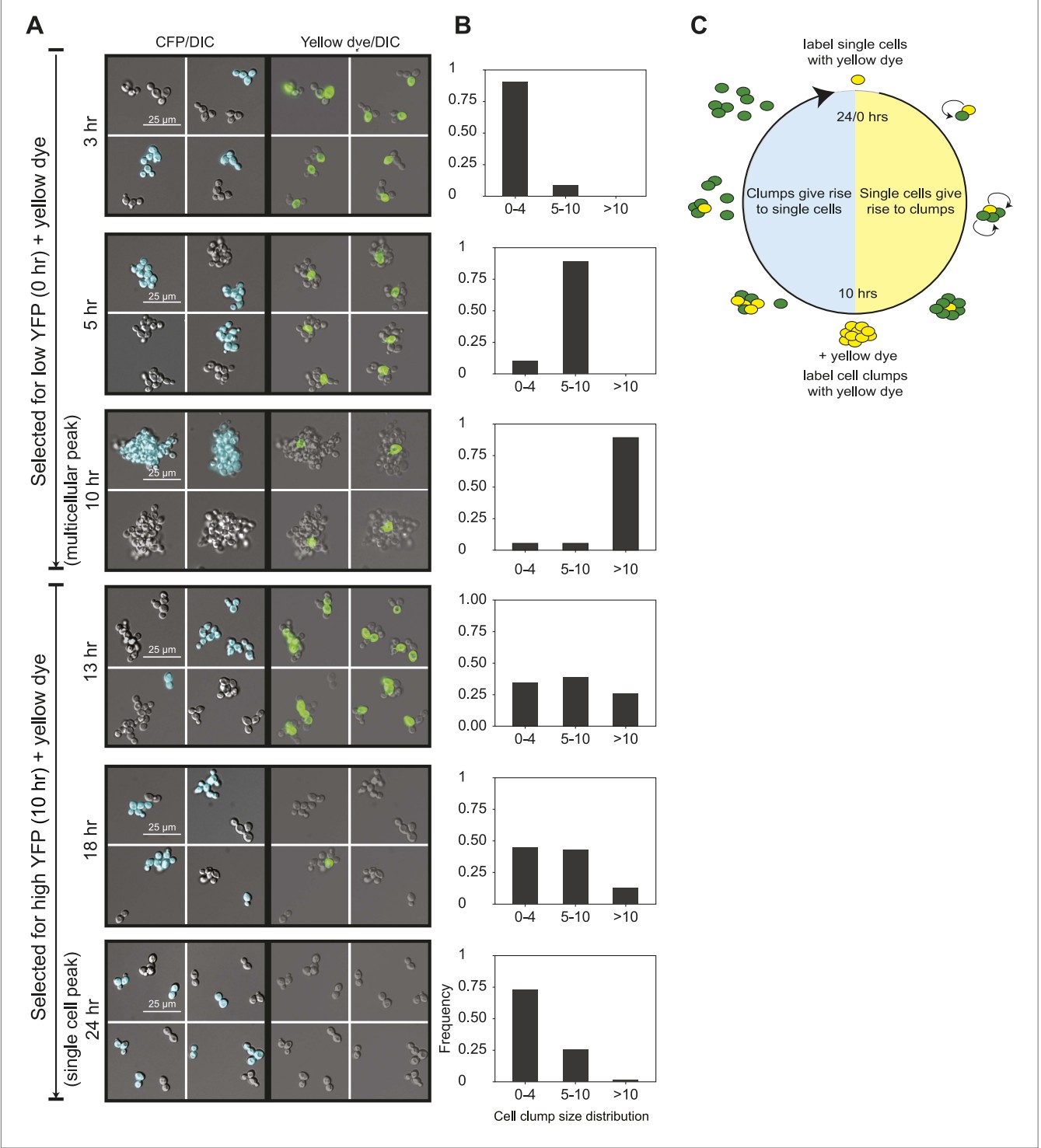

**Figure 2**. 24-hour autonomous oscillations in dynamic aggregate assembly. (**A**) Representative images of a synchronized E6-1 population at different points through the 24-hr cycle. The experiment contained two genetically similar subclones, one expressing cerulean fluorescent protein (CFP⁺) and the other not (CFP⁻). After FACS selection (0 and 10 hr) cell walls were covalently labeled with Oregon Green 488-x N-hydroxy succinimidyl ester, making them fluoresce brightly. Yellow-labeled single cells give rise to the multicellular phase by sticking to their daughters and clumps of different lineages do not mix (i.e., there are no CFP clumps with both CFP⁺ and CFP⁻ cells). Labeled clumps give rise to single cells and smaller clumps by clump fragmentation and producing single-celled daughters. Time scale is in reference to the initial collection of single cells (Low YFP selection, T = 0 hr) over a 24-hr period. (**B**) The population from the experiment described in (**A**) was quantified by calculating the frequency of clumps containing 1–4, 5–10, and >10 cells.
*Figure 2. Continued on next page*

*Figure 2. Continued*

There is a synchronous rise in clump size during the transition from single cells to clumps and a gradual decay of clump sizes during the transition from clumps to single cells. (**C**) Schematic representation of oscillator dynamics: single cells give rise to the multicellular phase by sticking to their progeny to form a lineage-based clump. At the peak multicellular phase, clumps give rise to the single cell phase by fragmentation and producing single-celled daughters some of which adhere to their progeny to produce small clumps.

The following figure supplement is available for figure 2:

**Figure supplement 1**. Dynamics of oscillator aggregate assembly.

peak aggregation gave rise to single cells by a combination of two mechanisms: fragmenting to produce smaller clumps and proliferating to produce single-celled offspring. Smaller clumps consisting of yellow cells and newly born non-yellow cells were observed 3 hr after (T = 13 hr) the peak aggregation time (10 hr). By 18 hr, the population was a mixture of medium-sized and smaller clumps and single cells, and by 24 hr non-yellow, single cells, and small clumps were the bulk of the population, representing the completion of one cycle. When unsynchronized CFP$^+$ and CFP$^-$ subclones were sonicated to break up all the clumps and cultured together for 90 min, clumps that re-associated were composed of a mixture of CFP$^+$ and CFP$^-$ cells showing that clumps form by cells sticking together rather than failing to separate after division (*Figure 2—figure supplement 1B*). These results show that cells oscillate between two states over a 24-hr period: single cells produce offspring that immediately stick to their mothers to assemble a lineage-based clump, which later produces a population of single cells and small clumps as a result of clumps fragmenting and producing single cells which divide to produce a mixture of small clumps and single cells (*Figure 2C*).

Because we typically sort populations twice every 24 hr, their behavior could reflect a damped oscillation, driven by selection, rather than a true oscillator, which should continue to cycle, even in the absence of selection. We therefore asked if our synchronized populations could free run: continue to oscillate in the absence of both entraining signals and periodic selection for cells in particular phases of the oscillation. We performed two experiments: one on populations and another to follow lineages initiating from single cells. In the population experiment, E6-1 was subjected to three different conditions. (1) Unsynchronized: cells that had been passaged without earlier synchronization were passed through the FACS machine and diluted twice every 24 hr, with intervals of 10 and 14 hr between the two dilutions, but all cells that passed through the FACS machine were collected, regardless of their fluorescence level and then diluted. (2) Synchronized: a population was synchronized by selection for 3 days with selection at alternating 10 and 14 hr intervals. The culture was synchronized by FACS for another 3 days, by FACS selection and dilution twice every 24 hr, with intervals of 10 and 14 hr between the two dilutions. (3) Free run: a population was synchronized by selection for 3 days at 10:14-hr intervals. After 3 days of synchronization (marked by a dotted line), this culture was not subject to FACs but was simply diluted twice every 24 hr, with intervals of 10 and 14 hr between the two dilutions for an additional 3 days. The initial, synchronized population used to start the free run was the same one that was used to start the population with continuing synchronization. Thus, all three conditions were diluted with intervals of 10 and 14 hr between the two daily dilutions and maintained at a density that allowed continual exponential proliferation. The free run populations continued to show detectable oscillations, which were undetectable in a control population that had not been synchronized (Unsynch) (*Figure 3A*). We next monitored how individual lineages oscillate without any prior FACS-based synchronization. A single cell was grown in a well of a 96-well plate and imaged every 3.5 hr. After 24 hr, the resulting population was collected and diluted back to single objects, regardless of clump size, into three separate new wells and followed every 3.5 hr for another 24 hr. This procedure was repeated one more time for each lineage (i.e., 3 cells from each of 3 sub-lineages) for a third day, as shown by the cartoon in *Figure 3B*. This protocol dilutes cells once every 24 hr to keep cells in exponential growth throughout the experiment. Cell lineages produced sustained oscillations for 3 days (*Figure 3C*) with many lineages remaining in phase with each other. In some lineages, their phase shifted within a 24-hr period, but the overall period, over 3 cycles was approximately 24 hr. These results suggest that we have evolved an autonomous oscillator capable of sustained oscillations in the absence of an environmental stimulus.

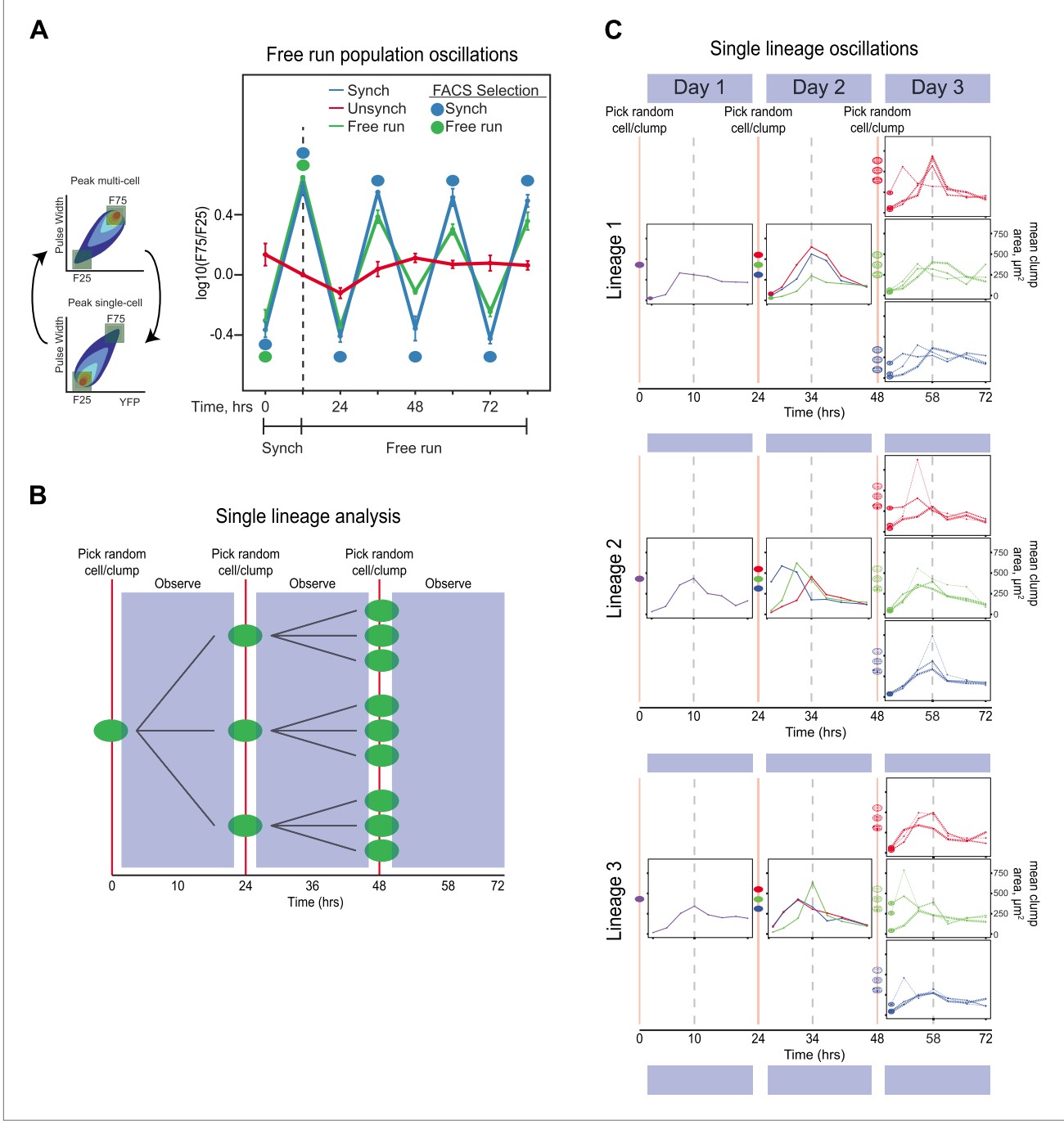

**Figure 3**. The evolved oscillator free runs for at least three cycles. (**A**) Three populations were compared. (1) Unsynchronized: cells that had been passaged without earlier synchronization were passed through the FACS machine and diluted twice every 24 hr, with intervals of 10 and 14 hr between the two dilutions, but all cells that passed through the FACS machine were collected, regardless of their fluorescence level and then diluted. (2) Synchronized: a population was synchronized by selection for 3 days with selection at alternating 10 and 14 hr intervals. The culture was synchronized by FACS for another 3 days, by FACS selection and dilution twice every 24 hr, with intervals of 10 and 14 hr between the two dilutions. (3) Free run: a population was synchronized by selection for 3 days at 10:14 hr intervals. After 3 days of synchronization (marked by a dotted line), this culture was not subject to FACs but was simply diluted twice every 24 hr, with intervals of 10 and 14 hr between the two dilutions for an additional 3 days. All populations were diluted to ensure the population always reached the same final, pre-dilution density of ~ $3 \times 10^5$ cells/ml, well below the density that corresponds to the end of mid-log phase growth for budding yeast ($3 \times 10^7$ cells/ml). The initial, synchronized population used to start the free run was the same one that was used to start the population with continuing synchronization (Synch). The measured variable, F75/F25 is based on the distribution of fluorescence and pulse widths at each time point. For both measurements, the lowest value is set to 0% and the highest to 100%, with values scaled linearly in between and the

*Figure 3. Continued on next page*

*Figure 3. Continued*

number of events that lie between 0 and 25% for both fluorescence intensity and pulse width is counted as F25, and the number of events that lie between 75 and 100% for both fluorescence intensity and pulse width is counted as F75, and we plot log(F75/F25). (**B**) Schematic of the protocol for analyzing individual lineages. A single cell is deposited in a microtiter well, and its progeny are observed every 3.5 hr for 24 hr, before taking 3 objects randomly (single cells or cell clumps), and depositing them in fresh wells, observing for 24 hr, and finally taking 3 objects randomly (cells or cell clumps), from each of the three wells and depositing them in fresh wells, and observing for 24 hr. (**C**) Individual traces of lineages originating from three separate cells (Lineage 1, 2, 3). An unsynchronized population was used to establish lineages that arose from single cells, which were diluted once every 24 hr and never experienced any FACS synchronization. A single cell was placed in 100 µl of medium in a microtiter well and allowed to proliferate for 24 hr that corresponds to a maximum of 16 divisions, a 64,000-fold increase in cell density from 10 cells/ml to 640,000 cells/ml, more than 40-fold below the density at which the exponential rate of cell proliferation starts to fall. Individual cells and clumps were imaged every 3.5 hr for 72 hr. At each time point, at least 50% of the cells and clumps in the wells were imaged and the average two-dimensional area was calculated. All lineages produce approximately 24-hr oscillations without any FACS-based synchronization. Some lineages alter their phase but continue to produce ~24-hr oscillations.

Because there could be multiple oscillating lineages, each with a different genetic basis, in our evolved population, we analyzed the behavior of four individual clones from the E6 population, focusing on the clone E6-1. We used bulk segregant analysis (*Yvert et al., 2003*; *Segre et al., 2006*; *Birkeland et al., 2010*; *Koschwanez et al., 2013*) to find candidates for the mutations that caused E6-1 to oscillate and verified them by engineering them into the ancestral, un-evolved strain. We began by crossing the evolved clone to its ancestor, putting the resulting diploid through meiosis, selecting for 24-hr oscillations on the resulting spores to isolate a pool of haploid progeny that showed robust cycling, and sequencing this pool's genomic DNA at high coverage. The full set of mutations needed to cause cycling should be present in the vast majority of the selected progeny, whereas the allele frequency of the neutral mutations that have accumulated during evolution should be ~50% (*Birkeland et al., 2010*; *Koschwanez et al., 2013*). We classified eight mutations that were present in ≥95% of the selected cells as putative causal mutations (*Figure 4A*). We recreated the evolved oscillator by engineering these eight mutations into a wild type laboratory strain. After synchronization, the oscillations of the recreated strain are indistinguishable from those of E6-1 (*Figure 4B*) and produce a similar free run behavior as E6-1 both as populations (*Figure 4C*) and as single lineages (*Figure 4D*).

We performed three tests to investigate the role of the different mutations in producing the evolved phenotype: deleting the mutated gene, replacing it with a wild-type allele, and investigating selection for or against the mutant allele when we selected for different levels of cell association. We investigated the role of removing genes because two of the causative mutations were stop codons, suggesting that other causative mutations might also inactivate the genes in which they occurred. Each causative mutation could be individually replaced by a gene deletion without destroying the oscillation (*Figure 5A*). For each of these single gene deletions, the corresponding wild-type version of the same gene was added back and tested for oscillations. Surprisingly, only two genes substantially altered the oscillations: *SIR4* made E6-1 single-celled and *YJL070C* produced constitutive clumps (*Figure 5B*). To determine how the mutations whose wild-type expression had no effect were contributing to the oscillator, pools of spores from a diploid heterozygous for the eight mutations were selected for four 'sub-traits' of the oscillator: batch culture was used to select for exponential growth and FACS to select for constitutive clumps, single cells, and stochastic clumping. After 5 days of selection, the eight mutations were PCR amplified and Sanger sequenced from genomic DNA of the surviving pool of spores to estimate the frequency of the ancestral and evolved allele (*Koschwanez et al., 2013*) (*Figure 5C*). In three independent replicate experiments, *sir4-100* was selected for in all conditions, *scw11-E261\** was selected for in constitutive clumps, and three mutations, *vps5-S520L*, *pet127-D650N*, and *whi2-R127\** were positively selected for in single cells and selected against in clumping cells. Different combinations of *HBN1*, *IDS2*, and *YJL070C* were selected for between replicate experiments in both clumping and single cell sub-traits, making it hard to assess their significance. Because of this variability, and the absence of a phenotype when *HBN1* or *IDS2* were either deleted or replaced with wild-type genes, it is unclear if these genes are required for the evolved oscillations, and if they are, how they contribute to them.

The nature of the mutated genes does not produce simple hypotheses for the mechanism of the oscillator: they have a wide range of molecular functions and no pair of them has a previously described functional relationship. In search of other clues, we measured gene expression to look for changes in E6-1. We performed two comparisons. The first was between synchronous and asynchronous

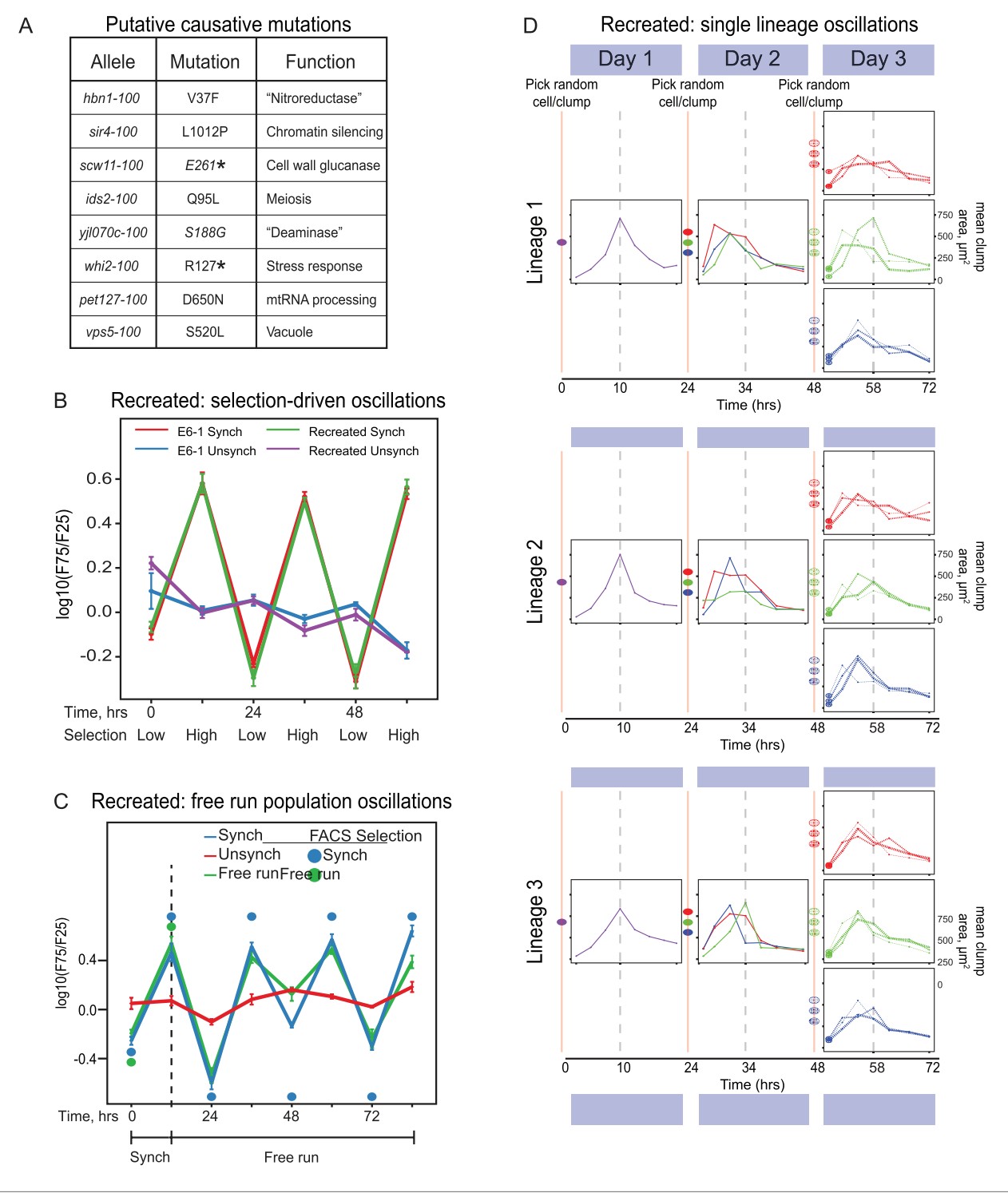

**Figure 4**. Reconstruction of an evolved oscillator (E6-1). (**A**) Table of putative causal mutations. The mutations lie in the coding regions of the genes, and the allele number indicates the amino acid substitution that results. An asterisk denotes creation of a stop codon before the midpoint of the open reading frame. (**B**) The eight putative causal mutations were engineered into a wild type laboratory strain (Recreated), which was synchronized and its oscillations were compared to the evolved E6-1 clone by plotting the log(F75/F25), as explained in the legend to **Figure 3A**. (**C**) The recreated strain shows a similar ability to oscillate without selection as the evolved clone E6-1. Both cultures were synchronized by FACS for 3 days (only the last cycle is shown) and then allowed to free run by growing them exponentially at a constant temperature in the dark. (**D**) Individual lineages produce 24-hr oscillations in the absence of FACS-synchronization when subjected to the protocol described in **Figure 3B**.

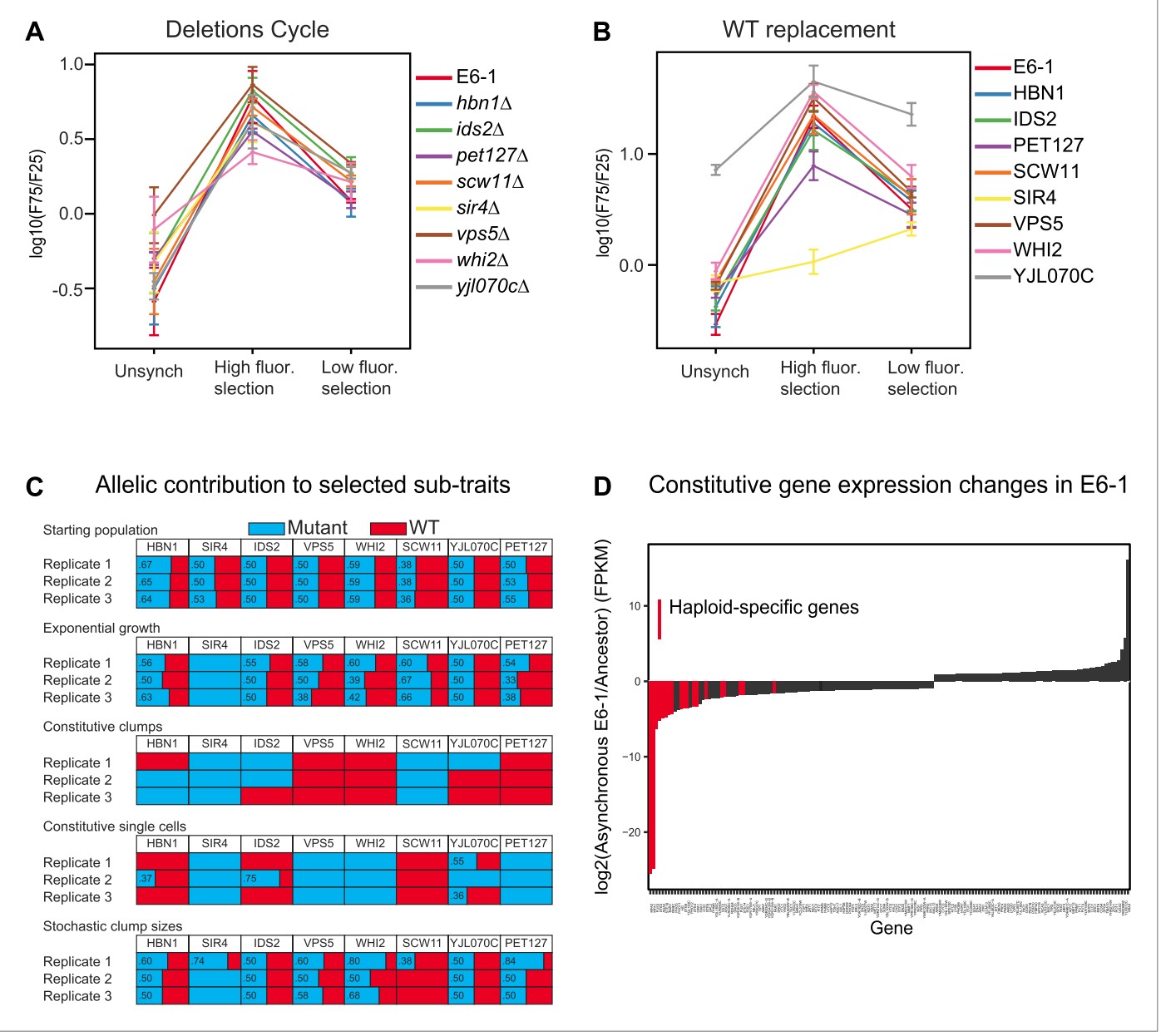

**Figure 5**. Analysis of putative causal mutations. (**A**) Genes containing causal mutations were individually deleted in E6-1 and tested for oscillations after applying the standard synchronization protocol to an unsynchronized culture. The metric for oscillation, F75/F25 is described in the legend to *Figure 3A*. (**B**) E6-1 strains containing a wild type copy (WT) of the indicated gene, at the *LEU2* locus, and deletion of the mutant allele from its endogenous locus were tested for oscillations after applying the standard synchronization protocol to an unsynchronized culture. (**C**) Effect of different mutant alleles on oscillator subtraits. The recreated strain was crossed to the ancestor to generate a pool of spores carrying all possible combinations of the wild type and evolved alleles at the genes that harbored putative causal mutations in clone E6-1. After selection, genomic DNA from the surviving spores was purified and the eight loci were PCR amplified and Sanger sequenced to estimate the relative frequency of the wild type and mutated alleles. (**D**) Gene expression changes in clone E6-1, relative to its ancestor. Six asynchronous E6-1 samples were analyzed as replicates against six aggregated ancestor samples. Haploid-specific genes (hsg), shown in red, are amongst the most repressed genes of E6-1. Overexpressed genes in the E6-1 are distributed among a variety of cellular processes. Data reported as the log2 ratio of E6-1 and ancestor fragments per kilobase per million fragments mapped (FPKM).

The following source data and figure supplement is available for figure 5:

**Source data 1**. Gene expression changes in clone E6-1, relative to its ancestor.

**Figure supplement 1**. Restoration of haploid gene expression abolishes E6-1 aggregates.

populations. We isolated RNA at the same times from three cultures: the peak periods of single cells and multicellular clumps for a synchronized E6-1 culture, an unsynchronized E6-1 culture passaged through the FACS machine, but without sorting, and an ancestral population subjected to the synchronization protocol. We compared each population to the other two and failed to find a significant excess of genes that appeared to cycle in the evolved, synchronized population.

The second comparison was to examine the mean expression of genes in the evolved and the ancestral populations. There were 155 genes whose expression differed between the ancestral and evolved populations (*Figure 5D* and *Figure 5—source data 1*). The most striking of these were genes involved in sexual behavior. Haploid-specific genes (hsg), which are normally expressed in haploids and repressed in diploids, were repressed in the evolved clone; 8 of the 10 most strongly repressed genes were haploid-specific. This phenotype is due to the causal mutation that inactivates *SIR4*, a gene required for transcriptional silencing of sub-telomeric genes, and the mating genes at the two silenced loci, *HMLα* and *HMR*a (*Rine and Herskowitz, 1987*; *Herskowitz, 1988*; *Pillus and Rine, 1989*). In *sir4* cells, the expression of a and α information from these two loci mimics a diploid cell and represses haploid-specific genes. Consistent with the effect of expressing *SIR4* in E6-1, deleting *HMLα* from the evolved clone, which removes the only source of α information restores the haploid pattern of gene expression and eliminates cell clumping (*Figure 5—figure supplement 1A,B*).

We investigated the evolutionary trajectory of the E6 population by asking when the causative mutations found in the E6-1 clone appeared and how widespread they were by the end of our evolution. Each mutation was PCR amplified and Sanger sequenced at different time points over the course of the evolution to determine its frequency to an accuracy of roughly 10%. None of the mutations in E6-1 reached a frequency above 0.4, two were undetectable in the overall population, and the remaining six were first detected between 12 and 30 days of evolution (*Figure 6A*). The low allele frequencies at the final time point of our evolution have two possible interpretations: the evolved population contains a single oscillating lineage, which has yet to eliminate a substantial fraction of non-oscillating lineages, or it is a mixture of genetically distinct oscillator lineages. All the clones from the evolved populations cycled suggesting that the populations have multiple lineages. We therefore performed bulk segregant analysis on three additional clones from the strongly selected population (E6-2,3,4) (*Figure 6B* and *Figure 6—source data 1A*). Our results show that two cycling populations have evolved entirely independent, and each of these has split into two subpopulations. E6-2 contained 6 of the 8 causative mutations found in E6-1 and had three additional putative causative mutations. E6-3 and E6-4 share three mutations, but have no mutations in common with the lineage that produced E6-1 and E6-2. Analyzing the entire open reading frame of *WHI2*, the one gene that was mutated in all 4 sequenced clones, suggests that there are at least five independently evolved oscillators in the E6 and E15 populations, since one population (E6) contains 2 mutant and one wild-type allele, and the other contains one mutant and one wild-type allele (*Figure 6—source data 1B*).

We asked whether the oscillations in other lineages also depend on the silencing of haploid specific genes. We measured mRNA levels of the two genes in *HMLα*, one of which, *HMLα2*, inhibits haploid-specific gene expression, and three representative haploid-specific genes in eight other clones derived from our two evolved populations (E6 and E15). All nine clones have increased expression of *HMLα* and reduced expression of haploid-specific genes when compared to the ancestor (*Figure 6C*). Furthermore, deleting *HMLα* in these clones resulted in a loss of multicellular aggregates in all but one clone (*Figure 6D*, E6-5), even though only one of the clones (E6-2) has the *sir4* allele found in E6-1.

## Discussion

Do the oscillations that we detect have any relationship to other oscillations that have been detected in budding yeast? Fluctuations in metabolic activity can be driven by temperature oscillations with a 24-hr period but rapidly die out at constant temperature (*Eelderink-Chen et al., 2010*), and a particular regime of growth to high density can induce prolonged metabolic oscillations whose period is 4–5 hr (*Tu et al., 2005*). We think it is unlikely that either of these cycles are connected to the behavior we observe: the 24-hr cycles do not free-run and the shorter cycles have the wrong period and can only be seen at cell densities higher than those that we used. We cannot, however, exclude the possibility that the behavior we see is the result of coupling changes in cell–cell adhesion to a previously unknown circadian clock in budding yeast.

How similar is the diurnal oscillator we evolved to naturally occurring circadian clocks? All circadian clocks share four features: (1) a limit-cycle, cell autonomous cellular oscillator that continues to

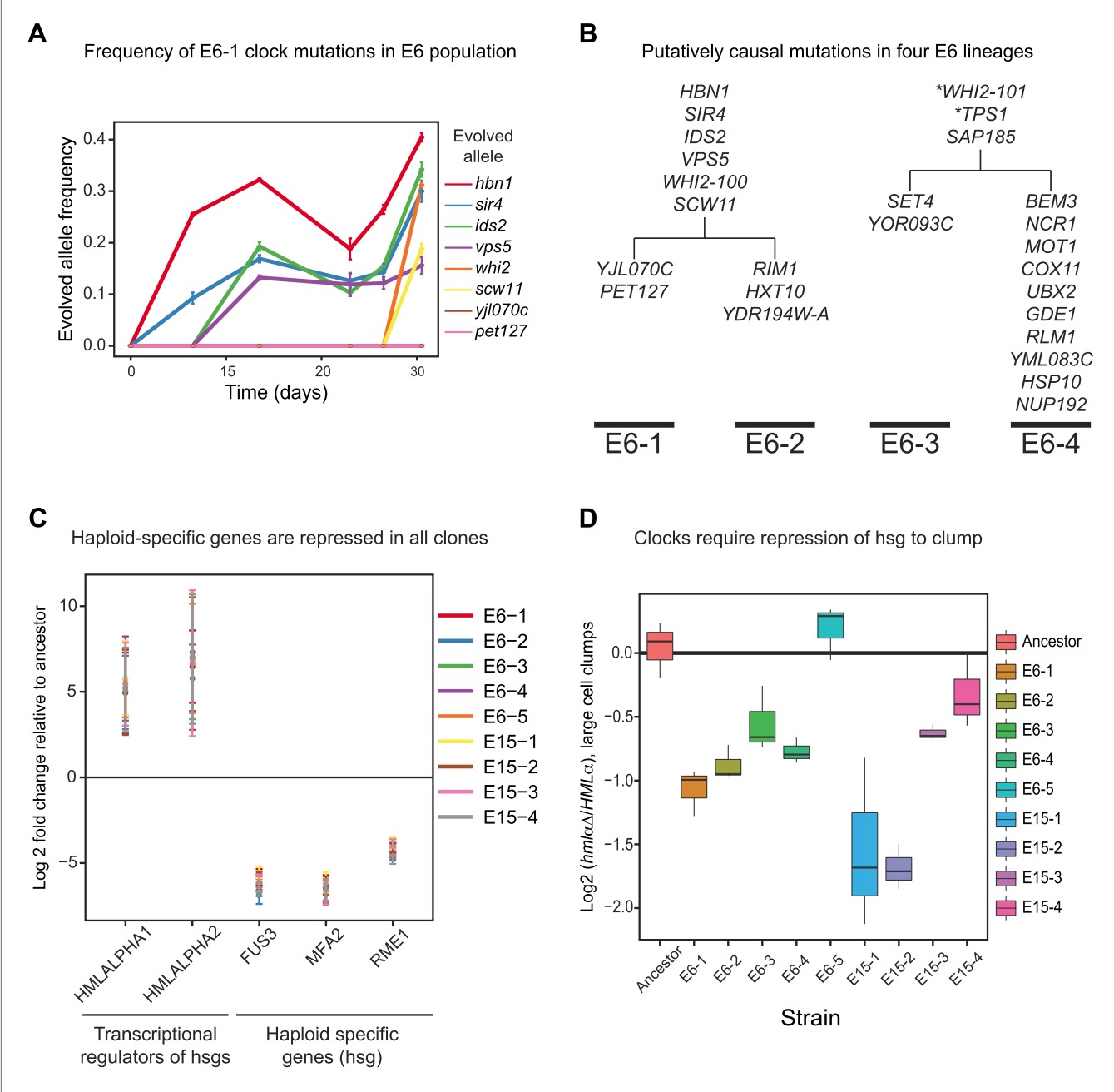

**Figure 6.** Genetically distinct clones evolve same oscillator through similar molecular changes. (**A**) The frequencies of the causal mutations found in E6-1 were measured by Sanger sequencing a region surrounding each mutation from frozen stocks of the E6 population taken at the indicated times during the experimental evolution. The frequencies of *yjl070c-S188G* and *pet127-D650N* were too low to measure by Sanger sequencing, whose detection threshold is an allele frequency of 10%. (**B**) Bulk segregant analysis and whole genome sequencing on three other isolates from the E6 population. Mutations that segregated at frequencies above 0.95 in at least one lineage are listed. All four clones contain a mutation in *WHI2*, but the mutation in E6-1 and E6-2 is a different allele from that in E6-3 and E6-4. Mutations with asterisks did not segregate above 95% in both lineages: *WHI2-101* segregated at 80% in E6-2 and *TPS1* segregated at 88% in E6-3. Given the read depth, the probability of producing these deviations from the expected frequency of a non-causal mutation (0.5) by chance is $6 \times 10^{-5}$ and $3 \times 10^{-8}$, respectively. (**C**) Quantitative PCR (qPCR) was used to compare the mRNA abundance of the two genes at *HMLα*, which encodes two regulators of mating type- and haploid-specific gene (hsg) expression, and three representative haploid-specific genes from eight clones (E6-1,2,3,4,5 and E15-1,2,3) to the ancestor. All eight clones have increased gene expression from *HMLα* and reduced haploid-specific gene expression. (**D**) The importance of repressing haploid-specific gene expression for the evolved phenotype was assessed by comparing cells that expressed both alleles of the mating type information (*HMLα*) to cells that expressed only *MATa* information (*hmlαΔ*).
*Figure 6. Continued on next page*

*Figure 6. Continued*

The frequency of clumps larger than 3 to 4 cells was compared in asynchronous cultures of *HMLα* and *hmlαΔ* derivatives of each clone. Four clones from E15 and three from E6 cannot form clumps when *HMLα* is deleted. E6-5 was unaffected by deletion of *HMLα*.

The following source data is available for figure 6:

**Source data 1**. Amino acid change of candidate causal mutations.

oscillate in the absence of environmental cues (free run); (2) the oscillator maintains a roughly 24-hr period over a range of temperatures (temperature compensation); (3) the period of the internal oscillator can be synchronized, or entrained, to the phase of a daily environmental cue (e.g., light, temperature); and (4) the ability to control cellular and organismal behavior (*Young and Kay, 2001*; *Tauber et al., 2004*; *Bell-Pedersen et al., 2005*; *Rosbash, 2009*). Our results show the yeast oscillator we evolved has two of these four behaviors: its roughly 24-hr period is maintained in the absence of environmental cues or selection on the output of the oscillator (*Figures 2B, 3A, 3C, 4C, and 4D*), and it regulates a cellular behavior, the ability of cells to form clumps. Further investigations are required to determine if the yeast oscillator uses the combination of positive and negative feedback loops that form the core of circadian clocks and could be evolved to be temperature-compensated and entrained by environmental signals.

Our experiment shows that mutating genes in multiple, unrelated biochemical networks can produce evolutionary novelty and reveals that identifying adaptive mutations is often insufficient to explain the mechanism of these new phenotypes. The ability of multiple combinations of mutations to produce the same complex phenotype leads to two inferences. First, we know relatively little about the circuitry of many existing molecular pathways and evolving novel traits may help illuminate previously unknown connections between pathways. Second, that multiple mutations are capable of producing unexpected changes in cellular physiology by connecting previously unrelated pathways to produce new circuits. The few mutations that are shared between different oscillating lineages suggest that there are multiple, functionally different ways that such rearrangements can lead to the same phenotype.

The common loss of silencing despite little commonality in other mutated genes between evolved clones has two implications. First, it suggests that wild yeast may regulate their clumpiness during the diploid phase, which accounts for most of their life cycle, and that our evolved populations have exploited this regulation to respond to our selection for oscillatory behavior. Second, the independent evolution of the same trait appears to have a mixture of common and diverse features. All nine of the populations we examined have switched from the haploid- to the diploid-specific pattern of gene expression and all four lineages analyzed in detail have mutated *WHI2*. The repeated occurrence of these genetic changes suggests that they greatly increase the number of subsequent mutations that could lead to the appearance of an oscillator. Whether the later mutations act on a single pathway or multiple, different pathways will require the detailed dissection of the oscillator's mechanism. It would be interesting to determine whether a selection for oscillation in diploids, where the inactivation of genes requires two independent genetic changes, would take a similar trajectory and produce the same mechanism that we have uncovered in haploids.

The inconsistency among bulk segregant analysis, gene deletions and wild type reversions illustrates the difficulty in identifying the contribution of different mutations to a complex, evolved phenotype. We can nevertheless make inferences about the molecular underpinnings of evolutionary novelties. Four of the six mutations that are strongly identified as causal mutations are loss of function mutations. The mutations in *SCW11* and *WHI2* are nonsense mutations, whereas those in *SIR4* and *YJL070C* can be replaced by gene deletions, but not by restoration of the wild-type gene. The preponderance of loss of function mutants suggests that sophisticated functions can be produced primarily by inactivating genes, rather than modifying or increasing their activities. Whether similar paths are followed in nature is likely to depend on the strength and unidirectionality of selection and the extent of antagonistic pleiotropy: the contrast between the benefits of a mutation in one environment and its costs in others. We speculate that short bursts of intense selection, such as those that occur when a species occupies a new and unoccupied niche, could lead to adaptation that is primarily driven by gene inactivation, whose short-term benefits exceeds its costs. These costs could later be reduced either by suppressor mutations that would minimize the harmful effects of loss of function mutations,

while maintaining the evolved phenotype, or even by reversion of the original mutation, once the evolved phenotype had been stabilized by further mutations. Further investigation will be required to determine how often loss of function mutations participate in a variety of evolutionary processes, ranging from cancer to the formation of new species and forms of biological organization.

## Materials and methods

### *S. cerevisiae* strains

The W303 *S.* cerevisiae strain background was used for all experiments. *Supplementary file 1* provides a detailed list of each strain used. Standard rich media (YPD) was used for all experiments: 2% Peptone, 2% D-Glucose, and 1% Yeast-Extract supplemented with Penicillin/Streptomycin (Sigma P0781).

### Flow cytometry

Analysis and sorting was performed on a MoFlo Legacy (Dako Cytomation/Beckman Coulter) with two excitation lasers and respective filter settings: 488 nm-550/30, and 594 nm-630/40. FACS FCS files were exported as tab delimited files using FlowJo software and plotted using R.

### Experimental evolution and clonal Isolation

Ancestral cells were inoculated into liquid media from a single colony and first grown for >12 hr in exponential phase at 30°C. At a fixed time of the day, cultures were centrifuged at 3000 rpm for 5–10 min and placed on ice. Cells were immediately brought to the FACS where ~5 × $10^5$ cells were collected from the bottom 6% or 15% of the YFP distribution. Data were collected from the first several thousand cells to set the gates for the selection that immediately followed. Cells were grown at 30°C in 40 ml YPD in beveled flasks for ~10 hr rotating at 110 rpm, centrifuged, and placed on ice for FACS selection. ~5 × $10^5$ cells were collected from the top 6% or 15% of the YFP distribution, grown at 30°C in beveled flasks containing 100 ml YPD for ~14 hr. This cycle was repeated for 30 continuous days. Cell densities were occasionally measured before FACS selection to ensure the population stayed below 1 × $10^6$ cells/ml. A small portion of the evolving population was occasionally frozen in 15% glycerol to establish a historical record. After 30 days of selection, clonal lines were isolated by spotting single events with FACS from the top 6% or 15% of the YFP distribution onto YPD plates. Resulting colonies were restreaked again on YPD to ensure the colonies were from a single cell.

### Synchronization of populations by FACS

For evolved populations, E6 and E15, a portion of the frozen stock was inoculated directly into liquid media and diluted to first grow exponentially overnight for >12 hr at 30°C. For clones, a single colony was inoculated into liquid media and diluted overnight for >12 hr, 30°C. At a fixed time of the day, which differed between different experiments, cells were centrifuged at 3000 rpm for 5–10 min, placed on ice, and immediately brought to the FACS machine. The dimmest 6–10% YFP expressing cells were collected from the unsynchronized population and grown in liquid media at <5 × 10^5 cells/ml for 10 hr at 30°C. Cells were then centrifuged, placed on ice, and brought to the FACS. Data were collected from the first several thousand cells to set the gates for the selection that immediately followed. The brightest 6–10% YFP cells were collected and grown at <5 × 10^5/ml for 14 hr at 30°C. Cells were prepared for FACS, data were collected while gates were set and the dimmest 6–10% YFP cells were collected to begin the next cycle. Unsynchronized populations were prepared for FACS in an identical manner to synchronized populations but cells were collected randomly from the entire YFP distribution. For free run experiments, populations were synchronized by FACS for 3 days and then split into two cultures. One culture was synchronized for another 3 days and the other was kept exponentially growing by transferring ~1 × $10^4$ cells into fresh media and analyzing the remaining cells by FACS every 10 and 14 hr for 3 days.

### Lineage oscillation free run

A time 0 hr, a single cell was deposited into a well of a 96-well plate containing 10 µl of YPD and immediately imaged with an inverted microscope. 190 µl of YPD was slowly added to the well, and the plate was placed at 30°C, unshaken. Every 3.5 hr, the plate was removed from the incubator, gently tapped by hand, quickly imaged, and placed back at 30°C. After 24 hr, the cells were collected and three cells regardless of their clump size were randomly chosen from the population by depositing every 200th FACS event from the entire YFP distribution into new individual wells containing 10 µl of YPD. Cells were

imaged and 190 μl of YPD was added to each well and placed at 30°C, unshaken and imaged every 3.5 hr as described above. After another 24 hr, the three populations were removed from the plate and three cells from each population were randomly deposited into new individual wells (9 total) using FACS as described above and imaged every 3.5 hr for 24 hr. For each time point, the average area of cell clumps was determined using ImageJ. A minimum of 250 cells were counted for each time point always making sure to count at least 50% of the population by choosing random fields of view to quantify.

### qPCR

qPCR was performed using published methods (*Koschwanez et al., 2013*).

### Quartile plots of oscillations

See *Figure 3A* for a definition of the F75:F25 ratio.

### Time course measurements of E6-1 clock mutations from the E6 population

A small fraction of the glycerol stock from each time point of the evolved population was collected and genomic DNA was immediately purified. A small region (~300 bp) surrounding each mutation was PCR amplified and Sanger sequenced. Frequencies were calculated as the ratio of the mutant and wild type peak heights on the sequencing electropherogram.

### Microscopy

Experimental procedures are described in the text. A description of the microscope and software used for acquisition is described elsewhere (*Koschwanez et al., 2013*). All images were modified for publication using imageJ.

### Cell wall dye

Cells were centrifuged, washed twice in PBS, and incubated with 100 μg of Oregon Green 488X succinimidyl ester (Life Technologies) in 250 μl of 0.1 M Sodium Bicarbonate in PBS for 3 to 5 min at room temperature. Dye was quenched with an excess of YPD, cells centrifuged, washed, and resuspended in YPD.

### Cell aggregate size comparisons

For *Figure 6D*, the ratio of cells above and below the pulse width (*Picot et al., 2012*) size of 14.5 μM polystyrene beads (Spherotech) was reported. To determine the clump size that corresponded to the pulse width of 14.5 μM beads, ancestor and E6-1 cells were collected by FACS onto a glass slide from a narrow gate that matched the 14.5 μM beads pulse width distribution. ~500 cells were scored for their clump size three separate times by eye with a microscope. The average clump size was the same for ancestor and E6-1 cells.

### Bulk segregant analysis

E6-1 was mated to yGW 556, a *MAT*α derivative of its ancestor and the resulting diploid was put through meiosis. The resulting spores were released from the ascus by treating the population with 50 μl of 2 mg/ml zymolyase (Zymo Research) diluted in water for 1 hr at 30°C. 450 μl of 1% Triton X-100 diluted in water was added and cells were sonicated for 10–20 s. Cells were centrifuged at 6000 rpm for 1 min and resuspended in YPD and grown overnight in exponential phase. $5 \times 10^5$ spores were selected in bulk for oscillations in the same manner E6 was originally evolved. Because one of the causal mutations eliminates silencing of *HML* and *HMR*, the spores that can cycle are incapable of mating. After 10 days of selection, genomic DNA was purified from the remaining pool of cells and sequenced at high coverage. The frequency of mutations in the mapped reads was scored using published methods (*Ares, 2012*; *Koschwanez et al., 2013*). Six putative causal mutations were identified and engineered into a wild-type strain but were insufficient to recreate the oscillator phenotype. This strain was mated to E6-1, sporulated, and individual spores were tested for oscillations. Equal amounts of 30 oscillating spores were pooled and genomic DNA was sequenced to identify the remaining two putative causal mutations in *SIR4* and *SCW11*.

### Subtrait selection

The recreated oscillator strain was mated to yGW 556, a *MAT*α derivative of its ancestor to produce a diploid heterozygous for the eight causal mutations. To generate a pool of spores that could not

mate during bulk selection and contained all possible combinations of E6-1 mutations, $P_{STE2}$-URA3, a marker that selects for *MATa* spores when plated on media lacking uracil and *LEU2-sir4-100*, a marker that selects for the *sir4-100* allele (which is sterile) when plated on media lacking leucine were introduced into the diploid strain. This strain was sporulated and the resulting spores were released from the ascus as described above and plated on either media lacking uracil or leucine. Equal numbers of cells from each media were pooled together, a fraction of this pool was frozen down to measure the initial frequency of each mutation (starting population) and the remaining cells were selected by FACS for the noted traits twice a day for 5 days. For exponential growth, cells were diluted by batch culture twice a day to keep them in exponential phase. After 5 days, genomic DNA was isolated from the pool of surviving spores, and the allele frequency of each mutation was determined by methods described above.

## Whole genome sequencing and RNA-sequencing

Genomic DNA (gDNA) was purified by resuspending >1 × 10⁸ cells in 50 μl of 0.5 M EDTA (pH = 7.5 or 8.0), 200 μl filtered water, and 2.5 μl of 20 mg/ml Zymolyase and incubating at 37°C for 1 hr. 50 μl of miniprep mix (0.20 M EDTA [pH = 8.0], 0.40 M Tris [pH = 8.0], 2% SDS) was added, mixed by inversion and incubated at 65°C for 30 min. 63 μl of 5 M KAc was added, mixed by inversion, and incubated on ice for 30 min. The cell lysate was centrifuged at 15K rpm for 10 min, and the supernatant was transferred to a new tube containing 720 μl of 100% Ethanol. gDNA was pelleted by centrifugation at 15K rpm and resuspended in 100 μl of water containing 1 μl of RNaseA (10 mg/ml) (Sigma) and incubated for 1 hr at 37°C. 2 μl of Proteinase K (20 mg/ml) (Sigma) was added and incubated for 2 hr at 37°C. 300 μl of isopropanol was added, and gDNA was pelleted by centrifugation at 15K rpm, washed in 80% Ethanol, air dried for 10 min, and resuspended in 10 mM Tris, pH 8.0. Total RNA was isolated following published protocols (*Ares, 2012*; *Koschwanez et al., 2013*). Genomic DNA and RNA sequencing libraries were made using Illumina Truseq DNA and RNA kits, respectively. Genome and mRNA sequencing was performed on an Illumina Hiseq 2000, reading 150 base pair paired-end reads.

## Acknowledgements

The authors thank Vlad Denic, Bodo Stern, Michael Desai, Diana Fusco, Matthias Kaiser, Fabio Zanini, and members of the Murray lab for helpful discussions and constructive criticism.

## Additional information

### Funding

| Funder | Grant reference | Author |
|---|---|---|
| National Institute of General Medical Sciences | Centers of Excellence in Complex Biomedical Systems Research P50 GM068763-09 | Andrew W Murray |
| National Institute of General Medical Sciences | NRSA Postdoctoral Fellowship 1F32GM085920-01 | Gregg A Wildenberg |

The funders had no role in study design, data collection and interpretation, or the decision to submit the work for publication.

### Author contributions

GAW, Conception and design, Acquisition of data, Analysis and interpretation of data, Drafting or revising the article; AWM, Conception and design, Analysis and interpretation of data, Drafting or revising the article

## Additional files

### Supplementary file

• Supplementary file 1. Table of the strains used and their relevant genotypes.

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
