## [Decision Letter]

Thank you for choosing to send your work entitled “Evolving a circadian oscillator in budding yeast” for consideration at *eLife*. Your full submission has been evaluated by Detlef Weigel (Senior editor) and 3 peer reviewers, one of whom, Michael Laub, is a member of our Board of Reviewing Editors, and one of whom, Michael Rosbash, has agreed to reveal his identity.

After careful consideration and discussion of your work, we have decided that we cannot consider it further for publication unless several issues can be resolved, through changes to the text and/or additional experiments. Most notably, there was uncertainty among the reviewers as to whether the evolved populations exhibited truly free-running oscillations. At the heart of this issue is Figure 1 showing YFP levels in several different populations over three days. The E6-1 line seems to oscillate between high and low YFP, but it was not clear whether this population was being subjected to FACS at 10 and 14 hour intervals or was simply being grown for 3 days and then analyzed periodically by flow cytometry. If it's the latter, then how long do oscillations persist? Related to this point of confusion, it was not clear why a synchronization process is required before performing the flow cytometry analysis in Figure 1: after 30 days of selection, shouldn't the evolved population already be synchronized? In this same vein, why doesn't the entire E6 population in Figure 1 show stronger oscillations? Although it is comprised of multiple clones, weren't they all oscillating by the end of the 30 day selection scheme that produced this population? And finally, there was a concern raised about whether the evolved population was oscillating or was being entrained by the selective regime imposed (FACS-based selection at 10 and 14 hour intervals).

There was general agreement that your work is exciting and creative, and it was considered potentially suitable for further consideration by *eLife*, provided you can adequately address the concerns summarized above. What would be most critical is a clear, unequivocal demonstration that the evolved population exhibits free running, circadian oscillations in the absence of any selection or specific growth procedure.

Further detail is provided below:

Reviewer #2:

In this interesting study, budding yeast was evolved to develop circadian rhythms; or something vaguely similar. The authors attempted to generate cycles of circadian gene expression, which ended up somewhat surprisingly selecting for daily cycles of cell-cell adhesion or clumping. There is no evident oscillating gene expression, which makes a marked distinction between this (these) evolved oscillator(s) and real circadian oscillators in different biological systems. The authors mapped their phenotype to eight mutations, all of which probably correspond to hypomorphic or amorphic alleles of the eight genes. At least they can be replaced by gene knock-outs. Finally, the authors demonstrate that independently evolved yeast clones have mutations in distinct sets of genes, which do not appear related in any obvious way. By this I mean the different mutant genes are not related to each other, nor is there a relationship between the different mutant genes in the different clones. An exception is the repression of haploid-specific gene expression, which is present in all but one of the clones. The paper in general is first rate; the experiments, figures and text are models of clarity. A key judgment call is significance. I come down positive despite my comments. Requests for experiments were kept minimal, in the spirit of *eLife* policy.

Major comments:

1) The authors might consider extending the issue of free-running oscillations as shown in Figure 1. How long do they persist in the absence of selection, is there a rapid loss of amplitude and what is their period? Are other clones similar to E6-1, which appears to have a fairly strong period that persists for at least two days? A period determination may require higher-density time-points.

2) Importantly, is the period temperature-compensated like all natural circadian oscillators?

3) I would replace individual mutant genes with wild-type versions, for at least a few of the mutant genes. The question is, are the oscillations completely eliminated by eliminating a single mutant gene, i.e., are all 8 mutations required for any perceptible cycling? I would begin this with Sir4, i.e., does this prediction hold based on what has been done? In other words, are oscillations eliminated for 7 of the 8 clones when WT Sir4 is restored, i.e., all but E6-5, as predicted from the hsg expression phenotypes? This would test the importance of mating type gene expression in a different way. This replacement of the mutant gene by a WT version should then be done for 2 or 3 more of the genes, in 2 or 3 more of the clones. The request is because it is unclear whether there are really 8 required mutations in all cases and whether this number or something like it is representative of all 8 clones.

4) Similarly, please check hsg in all 8 clones to see if this is really a common feature or not. This is especially important for clone E6-5, which did not respond to the deletion of HMLalpha. In other words, is it the only clone not to repress hsg? Check HMLalpha and HMLa expression too, to verify that this is truly the source of hsg regulation. Does upregulation of hsg expression itself inhibit clumping?

5) I would also check diploids with these 8 genes knocked out (both copies obviously). Do they undergo oscillations of adhesion? This is related to the statement:“[...]wild yeast may regulate their clumpiness during the diploid phase[...]” However, diploid yeast, although capable of filamentation, show very low levels of flocculation and expression of the flocculins. What about a direct assessment, if it is possible?

Reviewer #3:

I very much appreciate this paper and the creative nature of the science. I am bothered though by one potentially significant matter: evidence supporting the claim that the authors have evolved a circadian oscillator. There exists the possibility that the cells have adapted to the selective regime and that continuing to propagate the cells under the same 10 h and 14 h regime, even without FACS is at least in part responsible for the observed changes between single cells and clumps. I think that it is necessary to exclude this possibility. I do recognise that the authors report an experiment that attempts to rule out entrainment, but this still uses, from what I can, the 10 and 14 h regime. Perhaps take the cells and propagate them in a chemostat, or simply change duration of the cycle: take cells that are about to experience the 10 h phase and provide them with 14 h (and vice versa).

I would like to see the differences in intensity distributions after 30 days. I would also like to see the dynamics that take place during the synchronisation phase.

It wasn't clear to me how many cycles were used to synchronise the cells. It would be good to see the intensity distributions arising during the sync period. Actually this whole paragraph leaves a lot of detail unclear. For example, it is not stated that the purpose of the analysis is to determine whether oscillations in fluorescence continue following termination of selection (this is though mentioned in the caption to the figure).

It struck me as odd that after 72 hours and no FACS that the oscillations were still so regular. In the absence of any means to synchronisation I would expect to see regularity of oscillations decline. It makes me think there might be cues from the environment. See comment above. What happens if you continue to run through, say 10 days?

Analysis with OG is elegant. I take it though that the cells were always subjected to either 10 or 14 h regimes (and analysed at the end point). I am looking for an indication that clumping might be independent of culture conditions.

I found many aspects of the genetic analysis intriguing even if answers are few. The point that finding the mutations does not equate to doing genetics is well made. It seems quite remarkable that individual populations could contain independent oscillators. It suggests this phenotype is readily evolved and the possibility that yeast is already well set to switch between clumping and non-clumping states (the authors raise this possibility).

I felt the discussion of the genetic aspects to be less that compelling and rather too speculative; out in part out of kilter (context) with the rest of the paper.

[Editors' note: further revisions were requested prior to acceptance, as described below.]

Thank you for resubmitting your work entitled “Evolving a circadian oscillator in budding yeast” for further consideration at eLife. Your revised article has been favorably evaluated by Detlef Weigel (Senior editor), a Reviewing editor, and the original reviewers.

We had a long discussion among the two editors and the two reviewers how strong the evidence is that you have evolved de novo a circadian oscillator. In the end, we do not think that the evidence for a truly circadian, free running oscillator that reacts to entrainment and that is temperature compensated is all that strong. However, just evolving an oscillator is already an important finding that should be published in *eLife*. You should therefore remove the claim of a “circadian” oscillator from the title and the Abstract. You may discuss in the Introduction the properties of a circadian oscillator, and then develop in the Discussion what features of the de novo evolved oscillator possibly make it a diurnal, if not circadian oscillator.

In addition, the methodological details of the experiments done in Figure 3 to provide evidence for a free-running oscillator should be described in more detail in both the text and methods section. In particular, a clear description is needed of the growth and culturing conditions and how they differ (or not) from those used during the selection process.

---

## [Author Response]

*After careful consideration and discussion of your work, we have decided that we cannot consider it further for publication unless several issues can be resolved, through changes to the text and/or additional experiments. Most notably, there was uncertainty among the reviewers as to whether the evolved populations exhibited truly free-running oscillations*.

We now show the oscillator free-runs with two separate experiments one examining large populations of cells, and the other following individual lineages of cells.

*At the heart of this issue is*
Figure 1
*showing YFP levels in several different populations over three days. The E6-1 line seems to oscillate between high and low YFP, but it was not clear whether this population was being subjected to FACS at 10 and 14 hour intervals or was simply being grown for 3 days and then analyzed periodically by flow cytometry*.

We’ve included in the text a precise description of how and when populations are synchronized by FACS and present a clearer schematic of this process in Figure 2.

*If it's the latter, then how long do oscillations persist? Related to this point of confusion, it was not clear why a synchronization process is required before performing the flow cytometry analysis in*
Figure 1*: after 30 days of selection, shouldn't the evolved population already be synchronized?*

After 30 days, populations were frozen down in glycerol stocks for further analysis. Cells grown from these stocks are in different states of clumpiness and produce an asynchronous population and stay unsynchronized if passaged by batch dilution or passaged through the FACS machine without any deliberate selection for oscillating levels of YFP intensity.

*In this same vein, why doesn't the entire E6 population in*
Figure 1
*show stronger oscillations?*

There are two possible explanations, which are not mutually exclusive and both stem from the heterogeneity of the evolved population. The first is that lineages which do not oscillate or show irregular fluctuations in clump size may still be present because the lineages that produce regular oscillations have yet to entirely take over the population. The second is that the lineages that do oscillate could have subtly different periods (for example 22, 24, and 26 hours) and the E6 oscillation is a superposition of these different populations. Finally, as we state in the manuscript, the clone we chose to analyze, E6-1, is the clone that produces the strongest oscillations.

*Although it is comprised of multiple clones, weren't they all oscillating by the end of the 30 day selection scheme that produced this population? And finally, there was a concern raised about whether the evolved population was oscillating or was being entrained by the selective regime imposed (FACS-based selection at 10 and 14 hour intervals)*.

We have provided a clearer explanation of how the unsynchronized controls account for any affect the selective regime may have on the oscillations and we show in two different experiments that populations and individual cell lineages produce ∼24-hour free-running oscillations in the absence of the FACS selective regime.

Experiments we performed:

Free running oscillations. We asked whether cells oscillate in the absence of environmental cues and periodic selection in two ways: 1) populations of E6-1 and recreated cells were synchronized by FACS and then kept exponentially growing by batch culture for three days at a constant temperature (30°C) in the dark and we recorded distributions from a sample of the culture at 10 and 14 hour intervals. Figures 3 and 4 show that the clone E6-1 and the recreated populations can free run for at least three days. 2) We followed the behavior of individual lineages starting from a single cell by taking images with a microscope every 3.5hrs for 72 hours. Every 24 hours the population that grew from the single cell was diluted back to a single object (regardless of clump size) so that we could follow the behavior of individual lineages each circadian cycle (see Figure 3 for schematic of experiment). All cells continue to produce oscillations with an ∼24-hour period though some cells alter their phase relative to others (Figures 3 and 4).

Expression of wild type alleles in E6-1. We expressed a wild type copy of each of the eight mutated genes at the *LEU2* locus in an E6-1 strain deleted for the mutant copy of the same gene and tested for oscillations. To our surprise, only two strains had an obvious phenotype and the other six oscillated (Figure 5). To ask which mutations were necessary for the oscillator we selected for “sub-traits” of the oscillator from a pool of spores containing different combinations of the eight E6-1 mutations and asked which mutation segregated with each sub-trait (Figure 5). This provides evidence that five mutations contribute to the evolved phenotype. Thus we have evidence that mutations in six genes (one gene was implicated both by the phenotype of the wild type replacement and the segregation experiment) contribute to the evolved phenotype, but variability between replicate experiments makes it difficult to know whether mutations in *HBN1* and *IDS2* are required for oscillations.

Quantitative PCR (qPCR) of haploid-specific genes (hsgs). We measured the expression levels of the two known transcriptional regulators of haploid-specific genes (*MATα1* and *MATα2*) and three representative haploid-specific genes (*FUS3, MFA2*, and *RME1*) in the four other E6 clones and five E15 clones we reported on and found that the expression of the regulators rose and that of the haploid-specific genes fell in all of the clones we investigated (Figure 6).

Additional changes we have made:

We lengthened the manuscript and divided the data into six figures to fit the additional experiments and to make the text clearer.

We quantified the populations from the experiment in Figure 2 that initially only showed qualitative images of the population dynamics.

Reviewer #2:

*1) The authors might consider extending the issue of free-running oscillations as shown in*
Figure 1*. How long do they persist in the absence of selection, is there a rapid loss of amplitude and what is their period? Are other clones similar to E6-1, which appears to have a fairly strong period that persists for at least two days? A period determination may require higher-density time-points*.

We addressed this with experiments showing free-run oscillations in E6-1 and recreated strains.

2) Importantly, is the period temperature-compensated like all natural circadian oscillators?

We did not ask this question, which we argue lies outside the scope of this paper. Our primary focus was to determine how evolution can select for novel complex traits, regardless of how well they reproduce what has been evolved in nature. In organisms that cannot regulate their internal temperature, circadian oscillators will be selected for temperature compensation, but no such selection existed in our experiment, which was performed at a constant temperature.

3) I would replace individual mutant genes with wild-type versions, for at least a few of the mutant genes. The question is, are the oscillations completely eliminated by eliminating a single mutant gene, i.e., are all 8 mutations required for any perceptible cycling?

We performed this experiment and the results are discussed above (Figure 5).

*I would begin this with Sir4, i.e., does this prediction hold based on what has been done? In other words, are oscillations eliminated for 7 of the 8 clones when WT Sir4 is restored, i.e., all but E6-5, as predicted from the hsg expression phenotypes? This would test the importance of mating type gene expression in a different way*.

We only expressed *SIR4* in E6-1 because all of the other clones except E6-2 (a genetic relative of E6-1) are wild type for *SIR4*. Sir4 functions in a large complex with other proteins and the clones that are wild type for *SIR4* likely have a mutation in another gene with a role in silencing. The facts that deleting *HMLα* reduced clump size in every clone except E6-5, and haploid-specific genes were repressed in all the evolved clones argue very strongly that all the evolved lineages have acquired mutations that repress expression of haploid-specific genes, and that in all but one lineage, derepression of *HML*α has produced this change.

*This replacement of the mutant gene by a WT version should then be done for 2 or 3 more of the genes, in 2 or 3 more of the clones*.

For clone E6-1, we tested the phenotype of introducing the wild type versions of the genes that harbored the eight putative causative mutations, as discussed above.

Since most of these genes were not mutated in other lineages, they already contain the wild type version of the genes mutated in E6-1. Before we could perform the analogous experiment for other clones, we would first need to demonstrate that engineering the putative causative mutations into the ancestral strain would be sufficient to produce oscillations and this seems like an unreasonable amount of work to ask for.

*The request is because it is unclear whether there are really 8 required mutations in all cases and whether this number or something like it is representative of all 8 clones*.

This comment appears to contain two questions: 1) Does the oscillation of clone E6-1 really require the presence of all 8 putative causative mutations, and 2) do oscillations in other clones require similar numbers of mutations? On the first question, we present evidence that at least 6 of the 8 mutations in E6-1 contribute to the evolved phenotype. The bulk segregant analysis done on 3 other clones (Figure 6) identifies from 5 to 13 putative causative mutations.

4) Similarly, please check hsg in all 8 clones to see if this is really a common feature or not. This is especially important for clone E6-5, which did not respond to the deletion of HMLalpha. In other words, is it the only clone not to repress hsg? Check HMLalpha and HMLa expression too, to verify that this is truly the source of hsg regulation. Does upregulation of hsg expression itself inhibit clumping?

We checked expression of selected haploid-specific genes in all E6 and E15 derived clones. We show that deleting *HMLalpha* in E6-1 restores mating and reduces clump size, demonstrating that the expression of haploid-specific genes, which are required for mating, has been restored and that this restoration reduces clumping. A similar reduction in clump-size and restoration of mating was seen in six of the other seven evolved clones that we tested.

*5) I would also check diploids with these 8 genes knocked out (both copies obviously)*. *Do they undergo oscillations of adhesion? This is related to the statement:“[...]wild yeast may regulate their clumpiness during the diploid phase[...]” However, diploid yeast, although capable of filamentation, show very low levels of flocculation and expression of the flocculins. What about a direct assessment, if it is possible?*

We did not do this experiment because it would take too long to make the strains deleted for all eight genes. The reviewer is correct that diploids have different adhesion behaviors than haploids. The RNA-seq data and the effect of deleting *HMLα* shows that E6-1 shows the same pattern of gene expression (all haploid-specific genes are repressed, and candidate diploid specific genes are up) as bona fide *MAT*a/*MATα* diploids and thus argues that the experiment that the reviewer suggests is unnecessary.

Reviewer #3:

*I very much appreciate this paper and the creative nature of the science. I am bothered though by one potentially significant matter: evidence supporting the claim that the authors have evolved a circadian oscillator. There exists the possibility that the cells have adapted to the selective regime and that continuing to propagate the cells under the same 10 h and 14 h regime, even without FACS is at least in part responsible for the observed changes between single cells and clumps. I think that it is necessary to exclude this possibility. I do recognise that the authors report an experiment that attempts to rule out entrainment, but this still uses, from what I can, the 10 and 14 h regime. Perhaps take the cells and propagate them in a chemostat, or simply change duration of the cycle: take cells that are about to experience the 10 h phase and provide them with 14 h (and vice versa)*.

We performed experiments to show both evolved and recreated have a free run period without synchronizing selection.

*I would like to see the differences in intensity distributions after 30 days. I would also like to see the dynamics that take place during the synchronisation phase*.

We did not collect all of the intensity distributions during the experimental evolution and for those collected we had not yet developed a method to properly normalize them to allow accurate comparisons between measurements on different days. Figure 2 now has both qualitative images and quantification of the dynamics during the synchronization of cultures.

*It wasn't clear to me how many cycles were used to synchronise the cells. It would be good to see the intensity distributions arising during the sync period. Actually this whole paragraph leaves a lot of detail unclear. For example, it is not stated that the purpose of the analysis is to determine whether oscillations in fluorescence continue following termination of selection (this is though mentioned in the caption to the figure)*.

Figure 1 shows that E6 population fully synchronizes after two days, E15 takes three and the E6-1 clone synchronizes immediately following one round of FACS selection. We have rewritten this part of the text and provided better schematics to clarify how and for how long populations were subjected to FACS selection to synchronize them.

*It struck me as odd that after 72 hours and no FACS that the oscillations were still so regular. In the absence of any means to synchronisation I would expect to see regularity of oscillations decline. It makes me think there might be cues from the environment. See comment above*. *What happens if you continue to run through, say 10 days?*

We should have been clearer. In the original manuscript, there was no free-run experiment and we failed to make it clear that the populations that the reviewer comments on were still being synchronized. We now include a free-run experiment, as discussed above and have tried to make it clear when populations are being synchronized by twice a day FACS selection and when they are being allowed to free-run.

*Analysis with OG is elegant. I take it though that the cells were always subjected to either 10 or 14 h regimes (and analysed at the end point). I am looking for an indication that clumping might be independent of culture conditions*.

We are not sure what “OG” refers to. Because cultures are always kept at low density their growth does not cause changes in nutrient concentration. To eliminate the possibility that cells make compounds that modify the behavior of their relatives at a distance (quorum sensing) the experiments shown in Figure 2 were performed at very low cell densities (<10^3^ cells/ml), which are far below the densities at which bacterial quorum sensing occurs. We have even stronger evidence that the populations are responding to internal rather than external cues. We mixed single cells selected from the peak single cell phase and single cells selected from the peak multicellular phase, when the majority of cells are in clumps, with the two populations distinguished from each other by the expression of CFP. Examining this mixed population 10 hours later shows that the cells from single cell phase have produced substantially larger clumps than those from the multicellular phase, which produce a mixture of small and medium-sized clumps. This experiment demonstrates that cells from the different phases of the oscillator behave differently even when they share the same environment. Because the paper already has a number of complex experiments that are difficult to explain succinctly, our judgment was not to add this experiment to the revised manuscript.

*I found many aspects of the genetic analysis intriguing even if answers are few. The point that finding the mutations does not equate to doing genetics is well made. It seems quite remarkable that individual populations could contain independent oscillators. It suggests this phenotype is readily evolved and the possibility that yeast is already well set to switch between clumping and non-clumping states (the authors raise this possibility)*.

*I felt the discussion of the genetic aspects to be less that compelling and rather too speculative; out in part out of kilter (context) with the rest of the paper*.

We have addressed this concern with modifications to the text.